# Integrin α10-Antibodies Reduce Glioblastoma Tumor Growth and Cell Migration

**DOI:** 10.3390/cancers13051184

**Published:** 2021-03-09

**Authors:** Katarzyna Chmielarska Masoumi, Xiaoli Huang, Wondossen Sime, Anna Mirkov, Matilda Munksgaard Thorén, Ramin Massoumi, Evy Lundgren-Åkerlund

**Affiliations:** 1Xintela AB, Medicon Village, Scheeletorget 1, SE-223 81 Lund, Sweden; katarzyna@xintela.se (K.C.M.); xiaoli@xintela.se (X.H.); anna@xintela.se (A.M.); munksgaard83@gmail.com (M.M.T.); 2IVRS AB, Medicon Village, Scheeletorget 1, SE-223 81 Lund, Sweden; wondossen.sime@ivrs.se (W.S.); ramin.massoumi@ivrs.se (R.M.)

**Keywords:** function-blocking antibody, glioblastoma, integrin α10β1, collagen, migration, tumor growth

## Abstract

**Simple Summary:**

Glioblastoma (GB) is the most common and most deadly form of brain tumor in adults which currently lacks effective treatments. Thus, there is a high need to identify new and effective ways to target the aggressive GB cells and treat the GB patients. In this study, we investigated the treatment effect of two antibodies that have been developed to target the protein integrin α10β1, which is present on the surface of GB cells. Our results show that the growth of GB tumor cells is reduced in the presence of the α10β1 antibodies. The treatment effect is demonstrated both in cell experiments and in an animal model. In addition, we found that the antibodies reduce the migration of the GB cells. We suggest that function-blocking antibodies targeting the integrin α10β1 is a promising new approach to treat glioblastoma patients.

**Abstract:**

Glioblastoma (GB) is the most common and the most aggressive form of brain tumor in adults, which currently lacks efficient treatment strategies. In this study, we investigated the therapeutic effect of function-blocking antibodies targeting integrin α10β1 on patient-derived-GB cell lines in vitro and in vivo. The in vitro studies demonstrated significant inhibiting effects of the integrin α10 antibodies on the adhesion, migration, proliferation, and sphere formation of GB cells. In a xenograft mouse model, the effect of the antibodies on tumor growth was investigated in luciferase-labeled and subcutaneously implanted GB cells. As demonstrated by in vivo imaging analysis and caliper measurements, the integrin α10-antibodies significantly suppressed GB tumor growth compared to control antibodies. Immunohistochemical analysis of the GB tumors showed lower expression of the proliferation marker Ki67 and an increased expression of cleaved caspase-3 after treatment with integrin α10 antibodies, further supporting a therapeutic effect. Our results suggest that function-blocking antibody targeting integrin α10β1 is a promising therapeutic strategy for the treatment of glioblastoma.

## 1. Introduction

Glioblastoma (GB) is the most common and most aggressive malignant primary brain tumor accounting for 60–70% of all gliomas [1,2]. The prognosis of GB is very poor with a high degree of recurrence and a 5–year survival rate of 4–5%. Despite the improvement in chemotherapy and radiotherapy for GB, the current overall median survival for patients receiving therapy is still only around 1.5 years. Several factors may account for the challenge in developing effective drugs for GB, such as the infiltrating nature of GB, a lack of good preclinical models translating to humans, and blood–brain barrier impermeability that consequently decreases the effect of blood-borne therapeutic agents [3,4,5,6,7].

To date, only three drugs have been approved for the treatment of GB (www.fda.gov, 8 March 2021, [8]) one of which, Avastin, is a targeted antibody-based therapy for recurrent GB. Avastin binds to vascular endothelial growth factor (VEGF) and restricts the growth of new blood vessels required by the tumor for further development, but the effect of Avastin on GB is limited [9]. While a number of targeted therapies are under development for GB, several strategies, including checkpoint PD-1/PD-L1 inhibitors [10,11] and an antibody–drug conjugate (ADC) targeting epidermal growth factor receptor variant III (EGFRvIII) [12,13], have failed in late clinical studies. Thus, there is a need for new targets in the development of novel and effective glioblastoma therapies.

Integrins are cell surface receptors comprised of alpha (α) and beta (β) subunits that interact with various ligands either in the extracellular matrix (ECM) or on other cells [14,15]. Interactions between integrin receptors on tumor cells and ECM molecules have been shown to play a key role in critical tumor cell functions, such as cell migration, invasion, proliferation, and survival [15,16,17]. A number of integrins (e.g., α6β4, α5β1, αvβ6, αvβ3, αvβ5, and α7) are up-regulated in different tumor types, including GB and their expression correlates with poor patient survival [18,19,20,21]. We have recently shown that integrin α10β1 is highly expressed in GB tumor tissue samples and on isolated GB cells [22] and that an ADC targeting integrin α10β1 induce cell death of GB cells both in vitro and in vivo [22].

Integrin α10β1 was discovered by our group and originally identified as a type II collagen-binding receptor on chondrocytes [23]. We have reported that integrin α10β1 is also expressed by mesenchymal stem cells (MSCs) [24], and that integrin α10-selected MSCs have improved differentiation potential [25] and a therapeutic effect in an equine preclinical model of osteoarthritis [26]. Our previous work demonstrates that integrin α10β1 expression is mainly found in cartilage-containing tissues and some fibrous tissues, such as perichondrium, periosteum, and fascia surrounding muscle and tendon [27].

Although little is known about integrin α10β1 in cancer, recent studies suggest that it may play an important role in the development and progression of certain tumors. We have shown previously that integrin α10β1 is up-regulated in malignant melanoma cells compared to primary melanocytes, and associated with melanoma cell migration [28]. Okada et al. have reported that the growth and survival of myxofibrosarcoma cells are dependent on integrin α10 expression and that integrin α10 is regulated by Rac and mTOR signaling [29]. Integrin α10β1 has also been shown to be an important receptor on α-smooth muscle actin (α-SMA)-expressing stromal cells associated with human ovarian tumors in their binding to the HU177 cryptic collagen epitope [30]. Another study found that enhanced expression of integrin α10 was associated with tropism in the central nervous system (CNS), and showed a role of integrin α10β1 in CNS lymphomas [31]. These results, together with our findings that integrin α10β1 expression is restricted in normal tissues [22,27], support its use as a therapeutic target in certain cancers.

In this study, we investigated the therapeutic effect of function-blocking antibodies targeting integrin α10β1 on GB cells. We showed that two distinct integrin α10-antibodies inhibited the function of human GB cells in vitro and suppressed GB tumor growth in an animal model, supporting integrin α10β1 as a novel target in the treatment of GB.

## 2. Materials and Methods

### 2.1. Cell Lines and Reagents

Patient-derived glioblastoma cells were acquired from the Human Glioblastoma Cell Culture resource (HGCC; www.hgcc.se, 8 March 2021) at the Dept. of Immunology, Genetics and Pathology, Uppsala University (Uppsala, Sweden) [22,32]. The U3054MG, U3054MGLuc/GFP, U3046MG and U3078MG cells (all of mesenchymal subtype [32]) were cultured in DMEM/F12 w/Glutamax and Neurobasal media (1:1) (Gibco, Waltham, MA, USA), supplemented with B27 (Gibco), N2 (Gibco), 100 U/mL Antibiotic-Antimycotic (Gibco), fibroblast growth factor (FGF-2) (10 ng/mL) (Miltenyi, Bergisch Gladbach, Germany) and epidermal growth factor (EGF) (10 ng/mL) (Gibco) at 37 °C with 5% CO_2_ for both monolayer and sphere culture. All HGCC cell lines cultured as monolayer were expanded on Corning Primaria cell culture dishes coated with laminin-111 (1:100, L2020, Sigma Aldrich, St. Louis, MO, USA) and the sphere cultures were expanded in ultra-low attachment multiple 6-well plates. The GBM-37 cells were sampled under the biobank, “Södra sjukvårdsregionens CNS tumörbiobank’’ (Dnr 2018/37). GBM-37 cells monolayer culture was maintained in RPMI1640 (Gibco) supplemented with 10% FBS (Biological Industries, Beit HaEmek, Israel) and 100 U/mL Antibiotic-Antimycotic (Gibco). GBM-37 spheres cultures were expanded in ultra-low attachment multiple 6-well plates in medium containing DMEM/F12 w/Glutamax (Gibco), supplemented with B27 (Gibco), N2 (Gibco), 100 U/mL Antibiotic-Antimycotic (Gibco), fibroblast growth factor (FGF-2) (20 ng/mL) (Miltenyi), and epidermal growth factor (EGF) (20 ng/mL) (Gibco) at 37 °C with 5% CO_2_. The mouse myoblast cell line C2C12 transduced with integrin α10 vector (C2C12α10) or integrin α11 vector (C2C12α11) was cultured in Dulbecco’s Modified Eagle Medium (Gibco) supplemented with 10% FBS (Biological Industries) and 100 U/mL Antibiotic-Antimycotic (Gibco). The C2C12α10 and C2C12α11 cells were selected using G418 (1 µg/mL, Gibco) and puromycin (10 µg/mL, Gibco), respectively. To guarantee cell line authenticity, cell lines were aliquoted and banked, and cultures were grown and used for a limited number of passages before starting a new culture from stock. Cell lines were routinely tested for mycoplasma contamination. The human antibody Th101 and the mouse antibody mAbα10, raised against the human integrin α10 I-domain, as well as the control antibody human immunoglobulin G1 (IgG1) (Th301) were from Xintela AB, Lund, Sweden. The mouse control antibody IgG2a was from Innovagen AB, Lund Sweden.

### 2.2. Flow Cytometry Analysis

Prior to flow cytometry analysis, cells were incubated with primary antibodies for 30 min in the dark at 4 °C, washed twice with DPBS (Hyclone, SH3002802) containing 1% FBS and 0.1% sodium azide and incubated with a secondary antibody for 30 min in the dark at 4 °C. Flow cytometry analysis was performed using a BD Accuri C6 flow cytometer (BD Biosciences, San Jose, CA, USA).

### 2.3. Cell Adhesion

Cell culture plates (48-well) were coated with 0.1% gelatin (G1393, Sigma Aldrich) for 30 min at 37 °C or 10 µg/mL collagen type I from rat tail (C7661, Sigma Aldrich) overnight at 4 °C. Gelatin was diluted in distilled and sterilized water and collagen I was diluted in phosphate-buffered saline (PBS). Prior to the experiment, plates were incubated with 0.25% BSA (Sigma Aldrich) in PBS at 37 °C for 30–90 min to block non-specific binding. In parallel, the C2C12α10, U3054MG, and U3046MG cells were harvested, suspended into a single cell suspension in Hank’s Balanced Salt Solution (HBSS). The cells were pre-incubated for 30 min in the presence or in the absence of antibodies at a concentration of 10 µg/mL at 4 °C. The cells were then allowed to attach to the pre-coated plates for 40–60 min at 37 °C, and unattached cells were removed by washing with HBSS. Adherent cells were fixed with ethanol, stained with 0.09% crystal violet (Sigma Aldrich), and the absorbed dye was extracted by 10% acetic acid (Sigma Aldrich). The extraction was analyzed by measuring the optical density (OD) at 590 nm in the SpectraMax i3 multi-mode plate reader (Molecular Devices, San Jose, CA, USA). All the results were expressed as the percentage of adherent cells considering the untreated as 100%.

### 2.4. 3D Collagen Gel Migration

Collagen I gel solution (10 mL, final concentration 1.2 mg/mL) was prepared by mixing 7 mL of DMEM/F12 w/Glutamax medium and Neurobasal medium (1:1) supplemented with B27 and N2, 1 mL of 10x HEPES (0.2 M, pH 8.0), and 2 mL of collagen type I (6 mg/mL C2124, Sigma Aldrich), as described earlier [33,34]. All solutions were kept on ice before and during mixing. Collagen gels were prepared by adding the collagen I gel solution into a 96-well plate and incubating for 15 min at 37 °C. Spheroids pre-treated with 10 µg/mL of integrin α10 or control antibodies, were then placed into the collagen solution with a pipette. The collagen‒spheroid solution was allowed to polymerize for 60–90 min at 37 °C. After polymerization, 100 µL medium containing 10 µg/mL of the integrin α10 or control antibodies were added to each well to cause the collagen gel with spheroids to float in the antibody solutions. Cell migration from spheroids embedded in collagen gels was monitored under an inverted light microscope (EVOS M5000, Invitrogen, Life Technologies, Carlsbad, CA, USA) and photographed at different time points. The radial cell density profile of the spheroid was analyzed with the radial profile plug in Fiji to quantify the distance of migrated cells.

### 2.5. Monolayer Cell Proliferation

Cell-culture plates (96-well) were coated with 10 µg/mL collagen type I from rat tail (C7661, Sigma Aldrich) overnight at 4 °C. The C2C12α10 cells suspended in a culture medium containing 0.5% FBS were mixed with 5 µg/mL antibodies and seeded at a concentration of 10,000 cells per well in 100 µL. After 24 h of treatment, the bromodeoxyuridine (BrdU) (10 µM final concentration) was added and incubated for 2 h following the protocol of Cell Proliferation ELISA BrdU (Roche). Briefly, the cells were fixed and stained with an antibody against BrdU (anti-BrdU-POD). The cells were then washed with PBS and incubated with substrate solution for 5–30 min. The amount of BrdU incorporated into the cells was measured using a SpectraMax i3 multi-mode plate reader (Molecular Devices, San Jose, CA, USA) at a wavelength of 370 nm according to the manufacturer’s recommendation. Data were expressed as a percentage of viable cells, considering the control antibody-treated cells as 100%.

### 2.6. Sphere Cell Proliferation

Cells were seeded in 6-well, ultralow attachment plates (Corning #3471) to facilitate sphere formation. U3046MG and GBM-37 cells suspended in culture medium were mixed with 10 µg/mL of antibodies and seeded at a concentration of 30,000 cells per well in 1.8 mL and incubated for 8 days. Treatment was repeated four times (day 1, 3, 5, and 7) during the experiment. On day 7, BrdU was added at a final concentration of 10 µM and incubated with cells for 24 h. Spheres were collected and BrdU incorporation was detected using allophycocyanin (APC) BrdU Flow Kit (BD Pharmigen #51-9000019AK or #552598). Briefly, cells were permeabilized, fixed, and then treated with DNase in order to expose BrdU epitopes. The incorporated BrdU in newly synthesized DNA was labeled using APC fluorescent anti-BrdU antibody and quantified by fluorescent intensity of anti-BrdU APC using an Accuri C6 flow cytometer (BD Biosciences, San Jose, CA, USA).

### 2.7. Sphere Formation

Cells suspended in the medium were mixed with 10 µg/mL of antibodies and seeded in a 96-well, ultralow attachment, flat-bottomed plate (Corning #3474) at a concentration of 3000 cells in 100 µL medium per well, and cultured for 8 days. The antibody treatment was repeated four times during the experiment. Sphere formation was analyzed by counting the number of formed spheres using the ImageXpress Pico microscope (Molecular Devices, San Jose, CA, USA).

### 2.8. In Vivo Study

All in vivo experiments were performed in female NMRI-nu immunodeficient mice aged 6–8 weeks (*n* = 10 per group) purchased from Janvier Labs (France). Animal welfare and experimental procedures were carried out in accordance with international standards, and animals were maintained under specific pathogen-free (SPF) conditions. All experimental procedures were approved by the Malmö and Lund Animal Ethics Committee (Approval no. M123-14; Sweden). For the induction of tumors, mice were inoculated with 2 × 10^6^ U3054MGLuc/GFP cells in Matrigel (Corning) by subcutaneous injection into the right flank regions. Two weeks post-injection, the tumor growth was monitored by non-invasive 2D bioluminescence (BLI) imaging, using the In Vivo Imaging System (IVIS-CT) spectrum (PerkinElmer, Waltham, MA, USA). Mice showing tumor growth signals were randomized into respective treatment groups based on their average BLI signal intensity recorded in a defined region of interest (ROI) with average total flux (P/S) values.

The antibodies were injected intraperitoneally at a concentration of 5 mg/kg. Tumor growth was monitored using bioluminescence 2D- and 3D- uCT imaging. Briefly, mice were anesthetized with 3% isoflurane gas and injected intraperitoneally with 150 mg D-Luciferin/kg body weight in PBS prior to imaging. Acquisition of 2D images was taken sequentially with five intervals between different segments of exposures (Emission: open filter, f/stop: 1, binding: 8). The BLI signal intensity was quantified in total flux (P/S) after deducting the average background signal from the ROI measurement using the live image analysis software (PerkinElmer, Waltham, MA, USA). Tumor volumes were calculated using digital calipers based on the length and width of the tumors: [mm^3^] = (length [mm]) × (width [mm]) 2 × 0.5. (V = L × W2 × 0.5). The weight of the mice was recorded each week before the treatment with antibodies.

### 2.9. Immunofluorescence and Confocal Microscopy

Immunofluorescence labeling was performed in cryosections from mouse tumor tissues. Fresh frozen tumor tissues were embedded and frozen in optimal cutting temperature (OCT) compound (at approximately −60 °C). Frozen tissue was sectioned at 8 or 10 µm thickness and collected on SuperFrost slides (Thermo Scientific, Waltham, MA, USA). Slides were air-dried at 37 °C for 20 min, rinsed in PBS, and post-fixed in acetone (100%) for 5–10 min at −20 °C. Following rinses in PBS, sections were incubated in PBS-TX-BSA for 30 min at room temperature (RT).

Sections were incubated with primary rabbit antibodies made against the cytoplasmic domain of human integrin α10 (Xintela AB, Lund, Sweden) for 16 h at 4 °C. After rinses in PBS (3 × 3 min), sections were incubated with horseradish peroxidase (HRP)-conjugated secondary antibodies (goat anti-rabbit, DAKO Envision-HRP, Denmark) for 45 min at RT. Following rinses in PBS-TX (3 × 3 min) and once in PBS, sections were incubated in 4’, 6-diamidino-2-phenylindole (DAPI) solution for 15 min, rinsed in PBS, and mounted in “anti-fade solution” ProLong Gold (Invitrogen, Waltham, MA, USA). Immunofluorescence analyses were performed in a laser confocal scanning system (LSM 800 or 710, Zeiss, Germany) equipped for specific detection of the emission wavelengths for the used fluorophore.

### 2.10. Immunohistochemistry Staining

Tissues were isolated and mounted in OCT embedding compound to be fresh frozen at −60 °C in 2-methylbutane and dry ice. Mouse tumor tissue was sectioned (4 µm), collected on microscope slides (SuperFrost plus, Thermo Scientific), air-dried at RT for 30 min, and then stained with Hematoxilin, and antibodies to Ki67 (MA5-14520, Invitrogen, Waltham, MA, USA) and cleaved caspase-3 (ab2302, Abcam, Cambridge, UK). Sections were then re-hydrated and de-paraffinized by immersion in xylene (100%, 5 min × 2) followed by immersion in a graded alcohol series (100% ethanol 1 min × 2, 95% ethanol 1 min, 70% ethanol 1 min) ending with distilled water. Heat-induced antigen retrieval was performed in citrate buffer, pH 6.0, (Abcam) or TRIS-EDTA buffer, pH 9.0 (Abcam) for 10 min at 90 °C, followed by immersion in distilled water for 10 min and in PBS containing 0.05% Triton X-100 (Sigma-Aldrich, St. Louis, MO, USA, PBS-TX) (5 min × 2). Sections were then incubated in PBS containing 0.3% hydrogen peroxide for 10 min, followed by rinses in PBS (3 min × 3). Sections were incubated in protein blocking solution (ab64226, Abcam) for 10 min in RT and incubated either with antibodies to Ki67 (MA5-14520, Invitrogen) or to cleaved caspase-3 (ab2302, Abcam) for 16 h at 4 °C. After rinses in PBS (3 min × 3), sections were incubated with HRP-conjugated secondary antibody (goat anti-rabbit, ab205718, Abcam) for 30 min at RT. Following rinses in PBS (3 min × 3), sections were incubated 10 min in 3,3-di-aminobenzidine (DAB) substrate (ab64238, Abcam), rinsed in PBS (3 min × 3), and counterstained with hematoxylin (01820, Histolab, Gothenburg, Sweden). Sections were mounted in Pertex (00840-05, Histolab) and cover-slipped.

### 2.11. Quantification of Ki67 and Cleaved Caspase-3

The Ki67-labeled slides were scanned using a Hamamtsu S60 scanner with 40× resolution. The scanned images were annotated using Hamamatsu NDP.view2 (U12388-01) software selecting the viable central part of the tumor. The annotations were exported to individual images, each representing one or more parts of the tumor. The images were analyzed using ImageJ 1.53c. Each image was color de-convoluted using standard ImageJ function with the Hematoxylin DAB setting. A threshold value was set to identify the number of significantly labeled Ki67 cells, the total number of cell nuclei, and the tissue area. For cleaved caspase-3 quantification, a minimum of 20 images per group of labeled slides was analyzed using ImageJ 1.53c. In brief, the quantitation process consisted of the following steps. Each image was color de-convoluted using the standard ImageJ function with the Hematoxylin DAB setting. A threshold value was set to identify significantly labeled cells. The DAB-labeled area was captured as a percentage of the image size. Data were presented as the average percentage per group.

### 2.12. Statistical Analysis

The software GraphPad Prism 9.0 (San Diego, CA, USA) was used to generate figures and perform statistical analysis. Unless otherwise specified, unpaired two-tailed Student *t*-test was applied. Four levels of significance were used for a 95% confidence interval, where * *p* < 0.05; ** *p* < 0.01; *** *p* < 0.001; and **** *p* < 0.0001.

## 3. Results

### 3.1. Binding Specificity of the Integrin α10 Antibodies

In this study, we used human (Th101) and mouse (mAbα10) monoclonal antibodies raised against the I-domain of human integrin α10, and the respective control antibodies, human Th301 and mouse IgG2a. The specificity of Th101 and mAbα10 was investigated by flow cytometry using C2C12 cells (a mouse myoblast cell line) transfected with either human integrin α10 (C2C12α10) or human integrin α11 (C2C12α11). The results demonstrated that Th101 and mAbα10 specifically bind to C2C12α10 cells and do not cross-react with C2C12α11 cells (Figure 1A). The control antibodies did not bind to any of the cell lines (Figure 1A).The flow cytometry plots and the gatings are present in Figure 1B.

As shown in a competition binding assay, Th101 and mAbα10 recognize different epitopes on integrin α10 since the binding intensity of mAbα10 (3 μg/mL) to C2C12α10 was not affected by increasing concentrations of Th101 (0 to 9 μg/mL). Similarly, the binding intensity of Th101 (3 μg/mL) to C2C12α10 was not affected by increasing concentrations of mAbα10 (0 to 9 μg/mL) (Figure 1C).

### 3.2. Expression of Collagen-Binding Integrins on GB Cells

To investigate the function-blocking effect of the antibodies Th101 and mAbα10, we used the C2C12α10 cell line and four different GB cell lines, U3054MG, U3046MG, U3078MG, and GBM-37. Since expression of integrin α10β1 varies between different GB cell lines [22] and increases in 3D cultures compared to monolayer cultures, we first analyzed the expression level of integrin α10β1 on the four GB cell lines cultured in a monolayer or as spheres. We also compared the expression of integrin α10β1 with the other collagen-binding integrins α1β1, α2β1, and α11β1 to understand their possible role in the adhesion and migration assays.

We found that integrin α10β1 was low on U3054MG, and GBM-37 in monolayer cultures yet increased in sphere cultures. U3046MG showed high integrin α10 expression in monolayer cultures which further increased in spheres, while integrin α10 expression on U3078MG was intermediate and appeared to decrease in spheroid cultures (Figure 2A,B). All GB cells expressed integrin α2 to a high degree, whereas α1 was expressed at a significantly lower level besides GBM-37. α11β1 expression was low in all cell lines (Figure 2A,B). Furthermore, we investigated the expression of integrin α10 on U3054MG cells grown as a tumor subcutaneously in mice. By immunofluorescence analysis, we could confirm strong staining of integrin α10 in the tumor (Figure 2C), demonstrating that integrin α10 is up-regulated on U3054MG also in vivo.

### 3.3. Inhibition of GB Cell Adhesion and Migration

In cell adhesion experiments (Figure 3A–C), we found that Th101 significantly inhibited adhesion of C2C12α10 to collagen I-coated dishes (a 50% reduction), as well as the adhesion of U3054MG cells to gelatin-coated dishes (a 20% reduction), compared to the control antibody. This is similar to what we have previously shown with mAbα10 [22]. Using the GB cell line U3046MG, which has significantly higher integrin α10β1 expression compared to U3054MG, Th101 and mAbα10 inhibited adhesion by 60% and 90%, respectively, compared to the control antibodies (Figure 3C). All antibodies were used at a concentration of 10 µg/mL.

In migration experiments, U3078MG and GBM-37 cells, grown as spheres, were pretreated with 10 µg/mL of Th101, mAbα10, or control antibodies and placed in a well inside a collagen type I gel. To optimize the effect of the antibodies, an additional 10 µg/mL of the integrin α10 antibodies and the respective control antibodies were added to the wells to make the gels float in the antibody solutions. Migration of cells from spheres into the gel was then analyzed by microscopy after 24 h. The results showed that both Th101 and mAbα10 significantly reduced migration of U3078MG and GBM-37 cells in the collagen gel by 30–45% compared to the control antibodies (Figure 4A–C).

### 3.4. Inhibition of GB Cell Proliferation and Sphere Formation

To demonstrate the effect of the integrin α10 function-blocking antibodies on proliferation, first, we studied C2C12α10 cells grown as monolayer cultures in the presence of antibodies (10 µg/mL) for 24 h. As shown by BrdU incorporation analysis, Th101 and mAbα10 significantly inhibited cell growth by 60% and 65%, respectively, compared to control antibodies (Figure 5A).

We then investigated the effects of Th101 and mAbα10 on the proliferation of GB cells grown as spheres in the presence of 10 µg/mL of Th101, mAbα10, or respective control antibodies for 8 days. As shown by BrdU incorporation analysis, both Th101 and mAbα10 significantly inhibited proliferation of U3046MG and GBM-37 cells by 20–45% as compared to the control antibodies (Figure 5B,C).

In addition, we cultured U3046MG in the presence of 10 µg/mL of Th101, mAbα10, or respective control antibodies for 10 days, and then calculated the number of spheres that were at least 100 µm in diameter (Figure 5D). We found that Th101 and mAbα10 significantly inhibited the sphere formation of U3046MG cells by 24% and 26%, respectively compared to the control antibodies, which further supports the finding that Th101 and mAbα10 inhibit proliferation of GB cells.

### 3.5. Inhibition of GB Tumor Growth In Vivo

To evaluate the significance of integrin α10β1 on the progression of GB in vivo, we investigated the effects of the function-blocking antibodies Th101 and mAbα10 on tumor growth using the GB cell line U3054MG in a xenograft mouse model. The GB cells, labeled with luciferase, were implanted subcutaneously in the flank of nude mice (2 million cells/mouse). After 13 days, when a distinct tumor had formed, the mice were treated with intraperitoneal injections of Th101, mAbα10, and the respective human control antibody Th301 and mouse IgG2a (5 mg/kg in 200 µL PBS) twice per week (Figure 6A). Tumor growth was then monitored every week either by a bioluminescence imaging system (Figure 6B–D; Appendix A) or by using a caliper to measure the length and width of the tumor (Figure 6E).

The results show that both Th101 and mAbα10 treatment significantly inhibited tumor growth of the U3054MG cells by 70% and 56%, respectively, compared to the control antibodies. A significant difference by 50% and 30%, respectively, in tumor growth was seen already after two antibody treatments. The bioluminescence assay and the caliper measurement showed similar tumor growth curves and similar antibody effects (Figure 6C–E). In addition, representative Appendix A visualize the location of tumors from Th101, mAbα10, Th301, and IgG2a treated animals. Furthermore, by measuring the body weight of the antibody-treated mice weekly, we could conclude that the antibodies had no negative side effects on the treated mice (Figure 6F).

### 3.6. Suppression of Proliferation and Increase in Apoptosis of GB Cells In Vivo

We also performed immunohistochemical analysis of proliferation and apoptosis in tumor tissue samples obtained when the mice were sacrificed after 5 weeks. We found a tendency (not significant) of lower expression of the proliferation marker Ki67 in tumor tissues from mice treated with Th101 and mAbα10 by 84% and 51%, respectively, compared to mice treated with the control antibodies (Figure 7A,B). Furthermore, the number of cells expressing cleaved caspase-3, as judged by imaging analysis of the percentage of positive cell area, was significantly increased in tumors isolated from mice treated with Th101 and mAbα10 by 16% and 10%, respectively, compared to tumors from control-treated animals (Figure 7C,D). These results support the therapeutic effect of the functional blocking integrin α10 antibodies in the in vivo study.

## 4. Discussion

In recent years, numerous new treatments have been tested on patients with GB, including multiple cytotoxic chemotherapeutic agents, immune therapies, anti-angiogenic treatments, targeted therapies, and various combination regimens, with little or no improvement compared to the current Standard of Care thus far [10,35]. For example, immune checkpoint inhibitors based on monoclonal antibodies targeting PD-1 or cytotoxic T-lymphocyte antigen 4 (CTLA-4) have shown impressive results in the treatment of several types of cancer, but limited benefits so far in GB patients, which is mainly believed to be due to drug resistance [11]. Novel VEGF, as well as VEGF receptor inhibitors, are under clinical development for recurrent GB but have not demonstrated benefits in terms of survival outcome [10]. Poly (ADP-ribose) polymerase (PARP) inhibitors have become part of the standard treatment for several cancer types, yet little progress has been made as regards GB therapy [10,36]. Thus, the need for new therapeutic targets and new treatment approaches for GB patients is huge.

Our results in the present study suggest a novel therapeutic strategy for the treatment of GB with function-blocking antibodies targeting integrin α10β1. We found that treatment of human patient-derived GB cells with integrin α10-antibodies inhibits adhesion, migration, proliferation, sphere formation and suppresses tumor growth of the GB cells in a xenograft animal model.

We investigated two distinct monoclonal antibodies, a human antibody (Th101) and a murine antibody (mAbα10), both of which cross-react with human and mouse integrin subunit α10. Both specifically bind to integrin α10 subunit, but not to integrin α11, which is the integrin α subunit with the highest identity to integrin α10 [37]. We have also demonstrated here that the two integrin α10-antibodies bind to different epitopes on integrin α10. Thus, our finding that the two antibodies have similar inhibitory effects on the GB cells further validates integrin α10 as a therapeutic target in GB.

A function-blocking activity of both antibodies on GB cells was demonstrated in several cell assays including adhesion, migration, proliferation, and sphere formation. In these studies, we used four different GB cell lines including U3054MG, U3046MG, U3078MG, and GBM-37. We have previously shown that U3054MG, U3046MG, and U3078MG have varying expression levels of integrin α10 with the lowest expression on U3054MG in monolayer cultures [22]. This likely explains the lower effect of the antibodies on the adhesion of U3054MG compared to U3046MG in this study. However, we have shown that the expression of integrin α10β1 largely increases on U3054MG cells grown in spheres compared to monolayer cultures [22]. Here, we confirm these results and additionally show that integrin α10β1 is highly expressed on the U3054MG cells grown as a tumor in mice. The U3054MG cell line was established from a patient GB tumor that recurred after the patient had undergone surgery and radiotherapy (50.4 Gy in total), and is classified as a mesenchymal GB subtype [32] with high invasion capacity [38]. We, therefore, decided to use this aggressive cell line in the in vivo study.

The partial effect of the integrin α10-antibodies on adhesion and migration of the GB cells on collagen may also depend on the expression of the other collagen-binding integrins, α1β1, α2β1, and α11β1 [39,40]. Our results showed that integrin α2β1 is highly expressed on the four GB cell lines we used, while α1β1 was expressed at a lower degree and α11β1 only in minor amounts. Interestingly, in contrast to integrin α10, integrins α1 and α11 were not upregulated in spheroid cultures of the GB cells, suggesting a specific role for integrin α10 in tumor formation and growth. The expression of integrin α2 was consistently high also in spheroid cultures. Others have reported expression of integrin α2β1 on a subset of GB cells [41] and that an antibody to integrin α2β1 inhibited cell migration but not the proliferation of GB cells [42].

A number of other integrins have been reported to be expressed by GB cells and associated with the invasive phenotype of glioma cells including αvβ3, αvβ5, α3β1, α5β1, α6β1, and α9β1 [17,43,44,45,46,47]. Furthermore, antibodies directed to the integrin subunit αv [48] and to β1-integrins [49,50] have demonstrated an inhibitory effect on the growth of GB cells in vivo. However, it should be noted that the β1-subunit is common to several integrin receptors; thus, β1-integrin antibodies will target a number of integrins on both normal and GB cells.

In this study, we could demonstrate that two function-blocking antibodies, targeting the integrin α10β1, significantly reduced the growth of the GB cells U3054MG in a xenograft mouse model. The antibody treatment started two weeks after subcutaneous implantation of the GB cells and when a distinct tumor was detected by imaging analysis. We could detect a small but significant effect on tumor growth already after two treatments with the integrin α10 antibodies Th101 and mAbα10. After five weeks and eight treatments, Th101 and mAbα10 reduced tumor growth by 70% and 56%, respectively, compared to the control antibodies. Caliper measurements of the tumor volume demonstrated similar growth curves and antibody effects, confirming the imaging analysis results. The therapeutic effect of the integrin α10-antibodies was supported by reduced proliferation and increased cell death in tumors collected from treated animals.

Our results are in agreement with our ADC approach [22], where we demonstrated that an integrin α10 antibody conjugated with the cell toxin saporin induces cell death of GB cells both in vitro and in an orthotopic xenograft mouse model. In the previous study, we used the same GB cell line (U3054MG) but a different mouse monoclonal integrin α10 antibody that induced cell death by delivering the conjugated toxin to the GB cells [22].

The present study demonstrates that unconjugated integrin α10-antibodies with function-blocking activity have the capacity to reduce the growth of GB cells both in vitro and in vivo. These antibodies represent a different antibody modality with a different mechanism of action compared to the ADC. Future studies will reveal the preferred antibody modality targeting integrin α10β1 in the treatment of GB.

## 5. Conclusions

We have demonstrated for the first time that function-blocking integrin α10-antibodies inhibit GB tumor growth as well as the migration of GB cells. This further validates integrin α10β1 as a promising target in GB and suggest a novel therapeutic strategy for the treatment of GB and other high-grade gliomas using function-blocking antibodies targeting integrin α10β1.

## Figures and Tables

**Figure 1 cancers-13-01184-f001:**
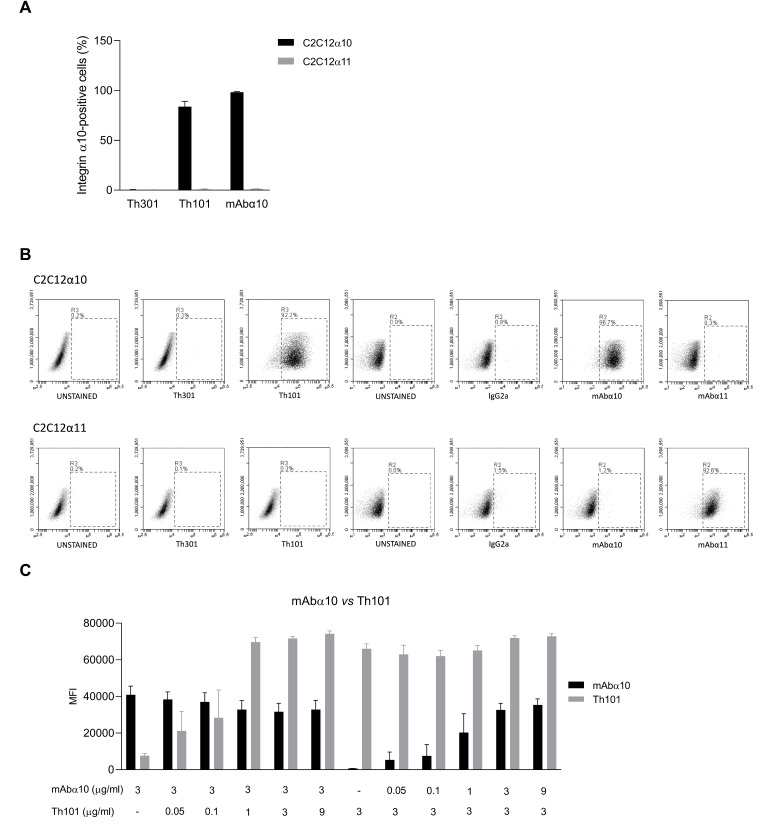
Integrin α10 antibodies bind specifically to the integrin α10 subunit. (**A**) Binding of the integrin α10 antibodies Th101 and mAbα10 and the human isotype control antibody Th301 to C2C12 cell lines overexpressing integrin α10 (C2C12α10) or integrin α11 (C2C12α11). (**B**) Specific binding of Th101 and mAbα10 to C2C12α10 cells. The gates were set to unstained cell samples. The figure shows cell plot shift when the C2C12α10 cells were stained with the integrin α10 human Th101 and mouse mAbα10 antibodies, but not with human Th301 and mouse immunogloblin G 2a (IgG2a) control antibodies or integrin α11 mouse monoclonal antibody (mAbα11). In C2C12α11 cells, there was a clear cell plot shift when the cells were stained with mAbα11, but not with Th101 and mAbα10. (**C**) Binding competition assay between the integrin α10 antibodies mAbα10 and Th101 using C2C12α10 cells. Data are expressed as mean fluorescence intensity (MFI) of 100,000 cells. Cells were incubated with antibodies at the indicated concentrations (μg/mL) and subsequently incubated with secondary antibodies. Immunolabeled cells were analyzed by flow cytometry. Data are compiled from at least three independent experiments as mean ± SE.

**Figure 2 cancers-13-01184-f002:**
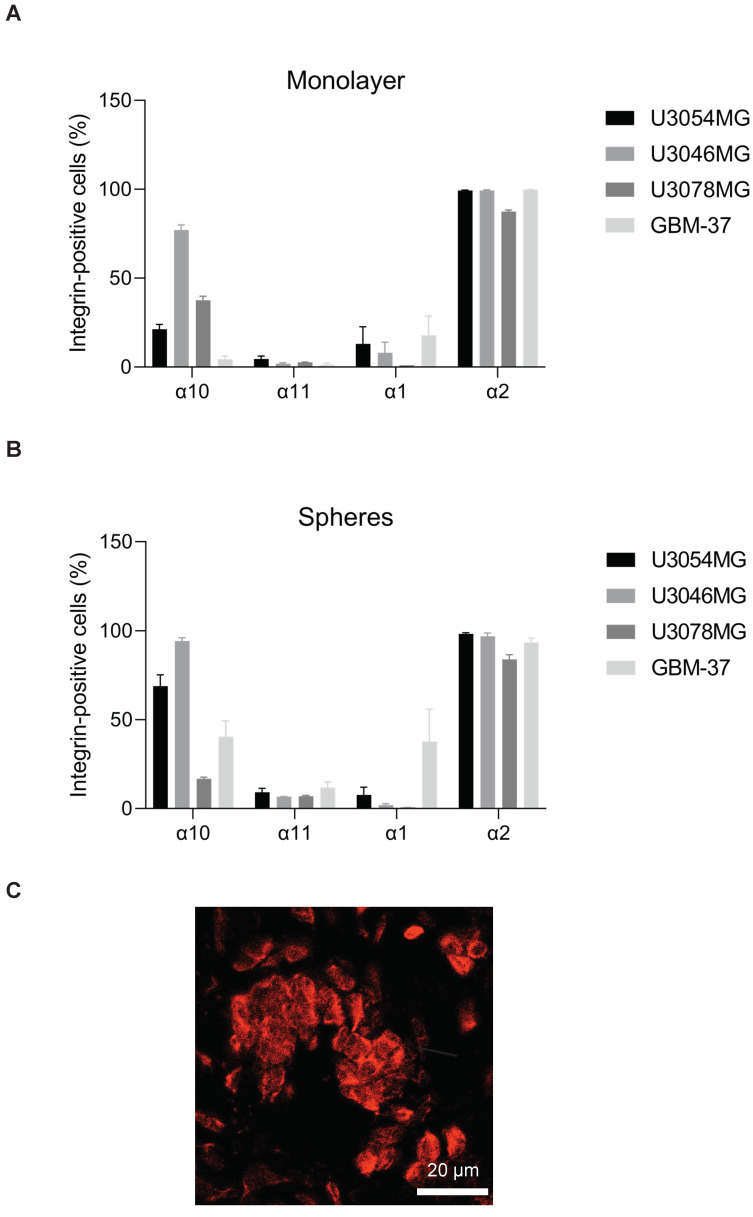
Glioblastoma (GB) spheroid cultures increase the expression of integrin α10β1. Flow cytometry analysis of the collagen-binding integrins subunits α10, α11, α1, and α2 on patient-derived GBM cell lines U3054MG, U3046MG, U3078MG, and GBM-37 grown either in monolayer (**A**) or in spheroid (**B**) cultures. The bar graphs show the percentage of integrin-positive cells compared to the whole cell population. (**C**) Representative immunofluorescent staining of integrin α10β1 in a GB tumor derived from U3054MG cells and grown in vivo. The scale bar represents 20 µm. Data are compiled from at least three independent experiments as mean ± SE.

**Figure 3 cancers-13-01184-f003:**
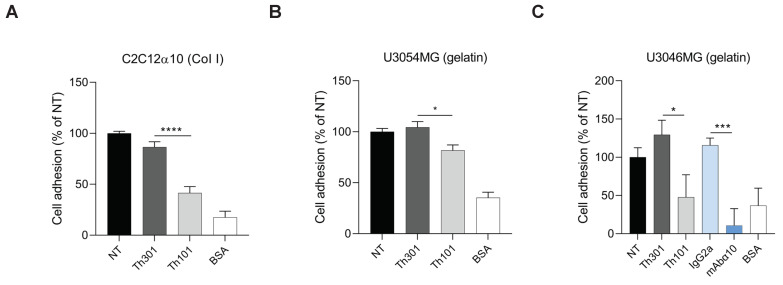
Integrin α10 antibodies inhibit GB cell adhesion to collagen and gelatin. Adhesion of C2C12α10 (**A**), U3054MG (**B**), and U3046MG (**C**) cells to dishes pre-coated with collagen I (Col I) or gelatin for 1 h in the absence or presence of integrin α10 antibodies (Th101 and mAbα10) or control antibodies (Th301 and IgG2a). Bovine serum albumin (BSA)-coated wells were used as a negative control. Adhered cells were quantified by crystal violet and spectrophotometric analysis. The bar graph shows the percentage of cell adhesion compared with non-treated (NT) cells. Data are compiled from three independent experiments as mean ± SE. Unpaired two-tailed Student *t*-test was applied. Four levels of significance were used for a 95% confidence interval, where * *p* < 0.05; *** *p* < 0.001; and **** *p* < 0.0001.

**Figure 4 cancers-13-01184-f004:**
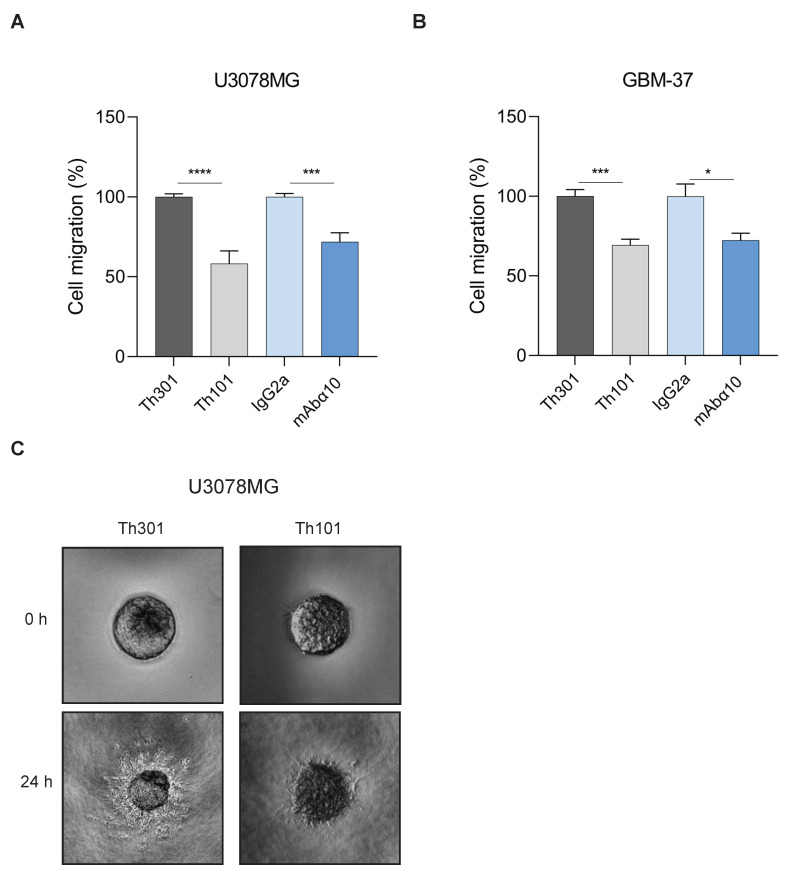
Integrin α10 antibodies inhibit GB cell migration. Spheroids composed of U3078MG (**A**) or GBM-37 (**B**) cells were embedded in collagen I gel and treated with integrin α10 monoclonal antibodies Th101 and mAbα10 or isotype control antibodies Th301 and IgG2a at 10 µg/mL. Spheroid migration was analyzed after 24 h. Relative migration was quantified as spheroid area difference (in pixels) at the indicated time points after subtraction of spheroid are at day 0. Data are expressed as mean ± SE of three independent experiments. (**C**) Representative pictures of U3078MG spheroids taken 24 h after embedding cells in the collagen gel (4× magnification). Unpaired two-tailed Student *t*-test was applied. Four levels of significance were used for a 95% confidence interval, where * *p* < 0.05; *** *p* < 0.001; and **** *p* < 0.0001.

**Figure 5 cancers-13-01184-f005:**
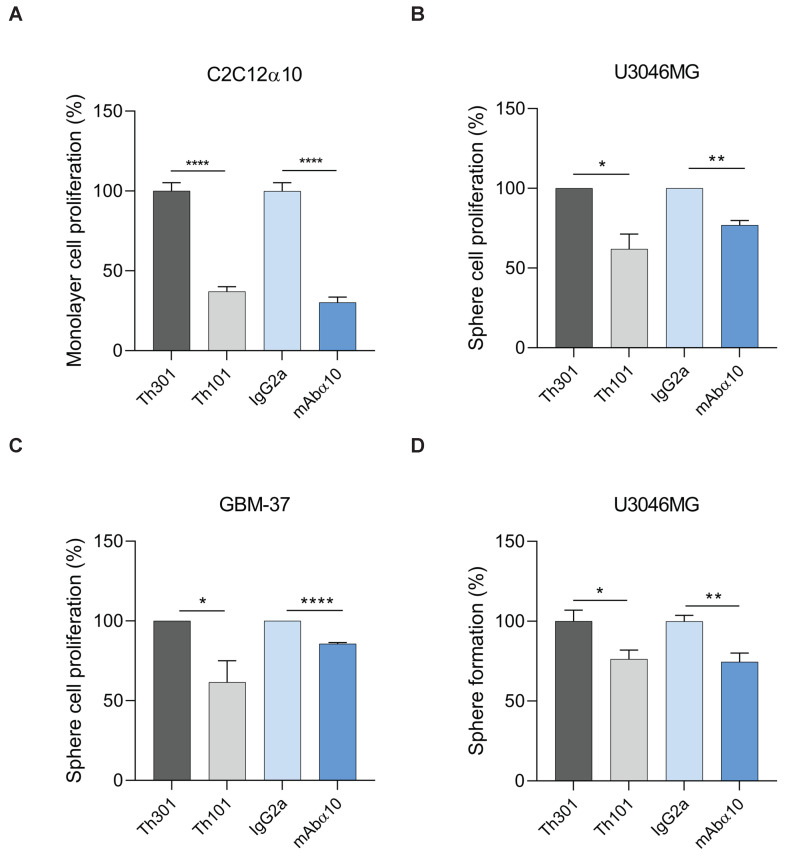
Integrin α10 antibodies inhibit GB cell proliferation and sphere formation (**A**) C2C12α10 cells were seeded in 96-well plates pre-coated with 20 µg/mL of collagen I (Col I). Cells were treated with Th101, mAbα10 antibodies or their isotype controls Th301 or IgG2a at 10 µg/mL concentration. After 24 h of incubation with the different antibodies, BrdU was added for incorporation into dividing cells during DNA synthesis and to detect proliferation. C2C12α10 cell proliferation was decreased in the presence of both human and mouse integrin α10 antibodies, but not by the control antibodies. (**B**,**C**) U3046MG and GBM-37 glioblastoma cells were seeded as sphere cultures and treated with 10 µg/mL of Th101, mAbα10, or control antibodies for 8 days. Twenty-four hours before harvest, BrdU was added. Cells were stained with APC-conjugated anti-BrdU antibody to measure the amount of BrdU incorporated in the replicating cells. The mean fluorescence intensity of BrdU was determined by flow cytometry and is shown as BrdU intensity of anti-integrin α10-treated cells normalized to its isotype control. (**D**) U3046MG cells were grown as spheres in the presence or absence of 10 µg/mL of Th101, mAbα10, Th301, or IgG2a. The number of spheres was analyzed on day 8 (mean ± SE). The number of spheres per field of microscope view in each experimental group was quantified by Fiji software and statistical analysis of the average diameter of spheres. All results were from at least three independent experiments with triplicates in each experiment and expressed as mean ± SE. Unpaired two-tailed Student *t*-test was applied. Four levels of significance were used for a 95% confidence interval, where * *p* < 0.05; ** *p* < 0.01; and **** *p* < 0.0001.

**Figure 6 cancers-13-01184-f006:**
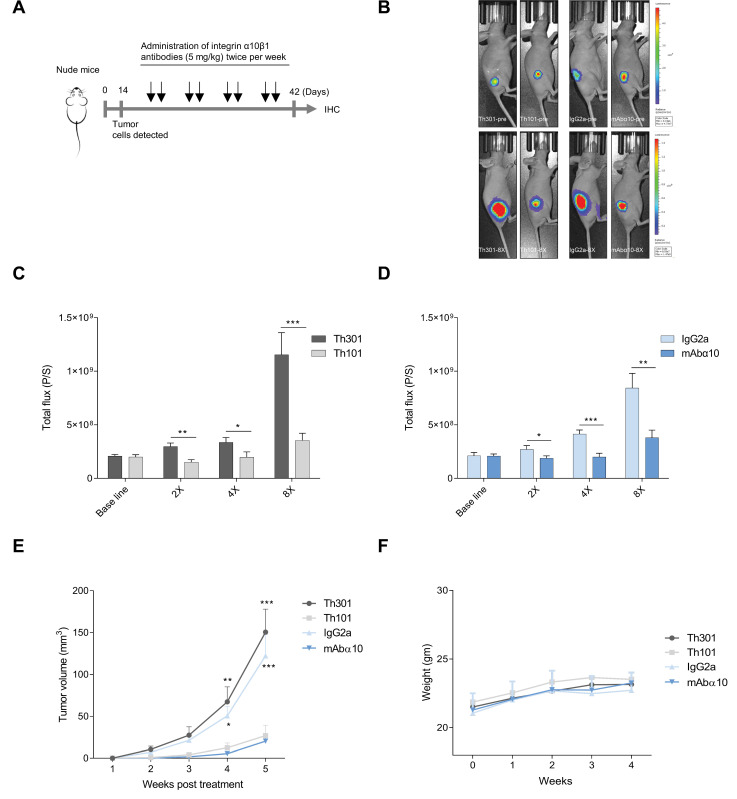
Anti-tumor efficacy of integrin α10 antibodies on xenograft glioblastoma in mice. (**A**) A schematic presentation of the in vivo experiment. (**B**) Photographic and 2D bioluminescence overlay images from representative mice from each treatment group before (Top) and after (Lower) treatment onset. The growth of U3054MG xenograft tumors treated with the integrin antibodies Th101 (**C**) and mAbα10 (**D**), and the control antibody Th301 and IgG2a. A total of 2 × 10^6^ U3054MG cells were injected subcutaneously into the right flank. After treatment onset, tumor progression in nude mice was monitored by luciferase live imaging weekly. The 2D bioluminescence signal measured as total flux (P/S) from each mouse was collected to evaluate tumor sizes of the Th101 and mAbα10-treatment groups as well as the Th301 and IgG2a-treatment groups. The data show the mean total flux (P/S) of all animals in each group at the indicated time point ± SE. The Mann–Whitney test was used to compare the difference between α10 antibody and control antibody and each treatment time point where * *p* < 0.05; ** *p* < 0.01; and *** *p* < 0.001. (**E**) The tumor size (mm^3^) in mice was measured by Vernier calipers during the treatment time. The two-way ANOVA test followed by Bonferroni’s multiple comparison test was applied. Four levels of significance were used for a 95% confidence interval, where * *p* < 0.05; ** *p* < 0.01; and *** *p* < 0.001. (**F**) Whole-body weight (gm) of the mice in each group measured over 4 weeks. Data are expressed as mean ± SE.

**Figure 7 cancers-13-01184-f007:**
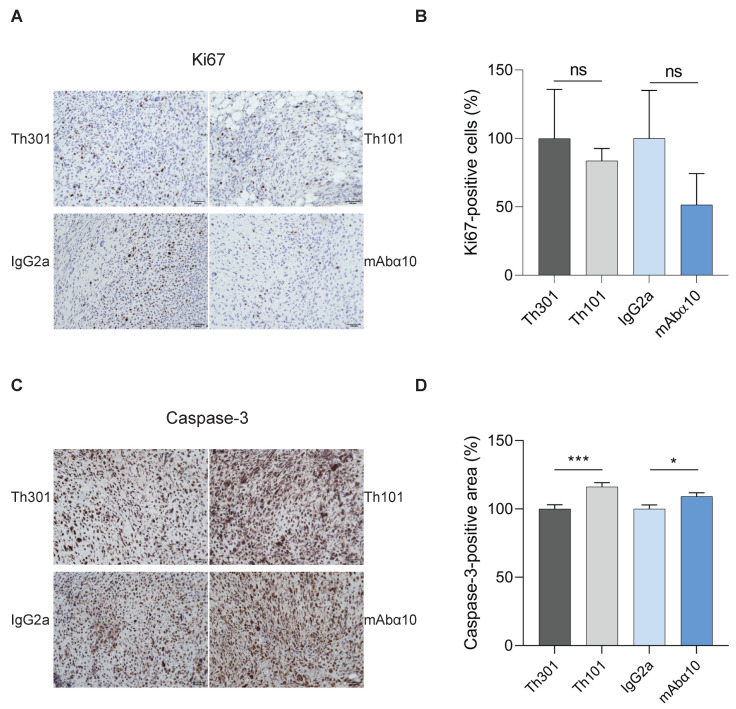
Integrin α10 antibody treatment reduces Ki67 and increases cleaved caspase-3 in GB tumors. (**A**) Representative immunohistochemical images of expression of Ki67 in the xenograft tumors from mice treated with control antibodies (Th301 and IgG2a) or integrin α10 antibodies (Th101 and mAbα10). Scale bars represent 10 µm. (**B**) The percentage of Ki67-positive cells in the tumors was reduced in Th101- and mAbα10-treated mice (*n* = 4 and *n* = 5, respectively). (**C**) Representative immunohistochemical images of cleaved caspase-3 expressions in xenograft tumors from mice treated with control antibodies (Th301 and IgG2a) or integrin α10 antibodies (Th101 and mAbα10). The scale bar represents 100 μm. (**D**) The percentage of cleaved caspase-3-positive cell area in the tumors was significantly increased in Th101- and mAbα10-treated mice compared to control mice. Results are presented as the mean ± SE from six tumor samples per group. Unpaired two-tailed Student *t*-test was applied. Four levels of significance were used for a 95% confidence interval, where * *p* < 0.05; *** *p* < 0.001; and ns: no significant.

## Data Availability

The data that support the findings of this study are available from the corresponding author upon reasonable request.

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
