# Peer review of "Integrin α10-Antibodies Reduce Glioblastoma Tumor Growth and Cell Migration"

_cancers, 2021, doi:10.3390/cancers13051184_

Round 1

Reviewer 1 Report

In this manuscript, the authors investigated the therapeutic effect of two distinct antibodies targeting integrin α10β1 on glioblastoma cells. They showed that these antibodies inhibited the function of human glioblastoma cells in vitro and reduced glioblastoma tumor growth in an animal model, supporting integrin α10β1 as a novel target in treatment of Glioblastoma.

 The authors came to very similar conclusion in a previous paper (Thoren et al., Cancers 2019) where they useda different integrin α10β1antibodyto target Glioblastoma cells. In that case, they conjugated the antibodyto the saporin, apotent cytotoxinand they demonstrated thatanti-α10-SAPinduced cell death of Glioblastomacells both in vitro and in vivo. Even if the approach and the strategy are not the same, I see a redundancy that somehow reduces the novelty of the paper submitted. I think the authors should discuss their choice of using two antibodies not conjugated to the drug and highlight the differences and t improvements of this new method, if any.

My Major concerns:

  • Authors should demonstrate the binding specificity of Th101 and mAbα10 antibodies using a western-blot assay. In fact, flowcytometry analysis cannot rule out cross reactivity while a western blotwill confirm that the antibodies are detecting the correct molecule.
  • Fig 1 and Fig 2: I would prefer to see data from all three experiments (including standard deviations) instead of a single representative experiment.
  • Why the authors did not choose to see the effects ofTh101 and mAbα10 antibodies in an orthotopic xenograft model as they did in Thoren et al., (Cancers 2019)? Orthotopic tumour models are more clinically relevant than their subcutaneous counterparts, due to the establishment of an organ-specific tumour microenvironment and allow researchers to examine cancer progression more accurately and interrogate drugs more faithfully.
  • Title: I suggest change the term “suppress” with “reduce”

Author Response

We want to thank the reviewer for the extensive review of our manuscript and the valuable and constructive suggestions.

Comments and Suggestions for Authors
In this manuscript, the authors investigated the therapeutic effect of two distinct antibodies targeting integrin α10β1 on glioblastoma cells. They showed that these antibodies inhibited the function of human glioblastoma cells in vitro and reduced glioblastoma tumor growth in an animal model, supporting integrin
α10β1 as a novel target in treatment of Glioblastoma.

The authors came to very similar conclusion in a previous paper (Thoren et al., Cancers 2019) where they used a different integrin α10β1antibodyto target Glioblastoma cells. In that case, they conjugated the antibody to the saporin, a potent cytotoxin and they demonstrated that anti-α10-SAPinduced cell death of
Glioblastoma cells both in vitro and in vivo. Even if the approach and the strategy are not the same, I see a redundancy that somehow reduces the novelty of the paper submitted. I think the authors should discuss their choice of using two antibodies not conjugated to the drug and highlight the differences and t
improvements of this new method, if any.

In the present study, we have demonstrated for the first time that function blocking integrin α10- antibodies inhibit migration and proliferation of glioblastoma (GB) cells and reduce GB tumor growth in an animal subcutaneous model. The results suggest that a function blocking antibody targeting integrin α10β1 is a promising therapeutic strategy in treatment of GB. We have previously demonstrated that an antibody-drug conjugate (ADC) targeting integrin α10β1 in an orthotopic animal model induce GB cell death in vitro and in vivo. Both studies suggest that integrin α10β1 is a novel and promising therapeutic target in GB, however, two different antibody modalities have been used and thus the results related to the antibody effect on the tumors and mechanism of action are different and complementary. Glioblastoma is a very aggressive tumor which is difficult to treat and new treatment strategies are highly needed. Although integrin α10β1 is a promising therapeutic target in GB, studies comparing two integrin α10 antibody modalities, a function blocking antibody versus an ADC, will provide important information regarding both safety, efficacy and mechanism of action and help to reveal which strategy is the most promising in the development of a safe and effective treatment for glioblastoma.

We used two specific antibodies binding to different epitopes on the integrin α10 subunit to investigate possible differences in their mechanism of action. The human antibody originates from a single chain antibody library and the mouse antibody by immunizing mice with the I-domain of integrin α10. Interestingly, our results demonstrated very similar blocking effect with the two antibodies both in vivo and in vitro in comparison with their respective control antibodies. The similar results further validate the role of integrin α10β1 as a therapeutic target and function blocking α10β1 antibody as a promising therapeutic strategy.

My Major concerns:
 Authors should demonstrate the binding specificity of Th101 and mAbα10 antibodies using a western-blot assay. In fact, flowcytometry analysis cannot rule out cross reactivity while a western blot will confirm that the antibodies are detecting the correct molecule.

Reply: We have addressed the specificity of the integrin α10 antibodies by using C2C12 cells transfected with either the human integrin α10 or integrin α11 subunits. The integrin α11 subunit is the integrin subunit with highest amino acid identity to integrin α10 and is thus a very good control to integrin α10. As we show in Fig 1, the integrin α10 antibodies show high binding to the C2C12α10 cells but not to C2C12α11. Another advantage with the C2C12 cells is that we analyse binding of the antibodies to the integrin α10β1 expressed in its natural conformation on the cell surface.

As many monoclonal antibodies, the antibodies we used in this study do not work in Western Blot. However, we have shown specific immunoprecipitation of the α10 and β1 chains using C2C12α10 and C2C12α11 cells and the same results are found with a polyclonal antibody directed to the cytoplasmic domain of the human integrin α10. This polyclonal antibody detects integrin α10 in Western blot. Finally, as we show in this study the integrin α10 inhibits adhesion of C2C12α10 but not C2C12α11 to their ligand collagen, which also demonstrate specificity.

Fig 1 and Fig 2: I would prefer to see data from all three experiments (including standard deviations) instead of a single representative experiment.

Reply: In the revised version of the manuscript, we have changed figures 1 and 2 which now show the results from at least three individual experiments.

Why the authors did not choose to see the effects ofTh101 and mAbα10 antibodies in an orthotopic xenograft model as they did in Thoren et al., (Cancers 2019)? Orthotopic tumour models are more clinically relevant than their subcutaneous counterparts, due to the establishment of an organ-specific tumour microenvironment and allow researchers to examine cancer progression
more accurately and interrogate drugs more faithfully.

Relpy: The aim of this study was to investigate the effect of function blocking integrin 10 antibodies on tumor growth by measuring the volume of the tumours and also investigate mechanism of action in in vitro assays. This is a different antibody modality compared to the previous orthotopic model where we used an ADC. We agree that orthotopic tumour models are more clinically relevant, however the subcutaneous model is good complementary model which add valuable information of the role of integrin α10β1 in tumor growth. In addition, the subcutaneous xenografts are easy to implant and the tumors are palpable and possible to measure.

 Title: I suggest change the term “suppress” with “reduce”

Reply: We changed the title of the manuscript to: Integrin α10-antibodies Reduce Glioblastoma Tumor Growth and Cell Migration

Reviewer 2 Report

In this study, Katarzyna Chmielarska Masoumi et al. suggest a novel therapeutic strategy for treating GB using function blocking antibodies targeting integrin α10β1 on patient derived-GB cell lines in vitro and in vivo.

The analyzed topic is of interest and the presented data are convincing, that if the Authors could address the points below reported, the paper might be proposed for publication in Cancers

Mayor revisions

Section Materials and Methods:

Paragraph 2.1. Cell Lines and Reagent – In this paragraph the cell culture medium for adherent cell lines isn’t reported, the Authors should insert this information.

Paragraph 2.4. 3 D Collagen Gel Migration - The Authors should insert bibliographic reference about the method described, clarify the time of pre-treatment of the spheroid with antibodies, that presumably are integrin α10 antibodies, and report the criteria for choosing the concentration of 10 µg/ml. Moreover, in this method and in the corresponding result (3.3. Inhibition of GB Cell Adhesion and Migration, lines 328-329), the Authors describe the addition of antibodies (20 μg/mL) “…to cause the collagen gel with spheroids to float” in the methods, and “….to cover the gel”, in the Results, however, the reason is unclear. They should specify, please.

Paragraph 2.6. BrdU Cell Cycle – The method reported is not for cell cycle analysis but for proliferation and is relative to the proliferation’ sphere and not adherent cell proliferation; the Authors should provide explanations.

Section Results:

In the figure 2C, the representative immunofluorescent staining of integrin α10 in a GB tumor derived from U3054MG cells and grown in vivo, should be moved in the description of the in vivo results, for greater uniformity.

The Authors should clarify and justify the choice of some specific cell types for some experiments and the other types for other analyses. For example, why have the Authors evaluated the proliferation in the U3046MG and GBM-37 spheres and not for the U3054MG sphere? Why have authors not analyzed the proliferation in the adherent cells?

The title of the 3.4 paragraph: Inhibition of GB Cell Proliferation and Sphere Formation, indicates the evaluation of proliferation of GB cell and not of sphere, but the data report the proliferation of spheres. Moreover two different treatment time are reported, 8 and 10  days. The Authors should clarify these observations.

Section Discussion:

In the Introduction or Discussion, the authors should comment on the reason for choosing to study the isoform α10 and not α2, which is highly expressed in all cell models.

In the Discussion, the Authors state: “However, we have shown that expression of integrin α10 largely increases on U3054MG cells grown in spheres compared to monolayer cultures [22]. Here, we confirm these results and additionally show that integrin α10 is highly expressed on the U3054MG cells grown as a tumor in mice. The U3054MG cell line was established from a patient GB tumor that recurred after the patient had undergone surgery and radiotherapy (50.4 Gy in total),and is classified as a mesenchymal GB subtype [32] with high invasion capacity [36]. We therefore decided to use this aggressive cell line in the in vivo study.”

In my opinion these reasons are not completely accurate. In fact, the Authors didn’t use spheres to induce glioblastoma in vivo, we do not know if using the U3046, the expression of α10 in vivo would have been as high as with the U3054. Moreover, the Authors should specify the molecular subtype of U3046.

Finally, the authors should comment on whether the evaluation of α10 expression in the patients' tumor could guide the choice of therapy. The choice of the U3054 line would make this opportunity fall.

Minor revisions

Do the terms α10 and α10β1 indicate the same integrin? The authors should clarify this doubt, and if these terms express the same molecule, the authors should ever use the single way.

Author Response

Response to Reviewer 2 Comments

We want to thank the reviewer for the extensive review of our manuscript and the valuable and constructive suggestions.

Comments and Suggestions for Authors
In this study, Katarzyna Chmielarska Masoumi et al. suggest a novel therapeutic strategy for treating GB using function blocking antibodies targeting integrin α10β1 on patient derived-GB cell lines in vitro and in vivo.
The analyzed topic is of interest and the presented data are convincing, that if the Authors could address the points below reported, the paper might be proposed for publication in Cancers

Mayor revisions

Section Materials and Methods:

Paragraph 2.1. Cell Lines and Reagent – In this paragraph the cell culture medium for adherent cell lines isn’t reported, the Authors should insert this information.

Reply: Information regarding cell culture medium has been included in Materials and Methods under Cell Lines and Reagents.

Paragraph 2.4. 3 D Collagen Gel Migration - The Authors should insert bibliographic reference about the method described, clarify the time of pre-treatment of the spheroid with antibodies, that presumably are integrin α10 antibodies, and report the criteria for choosing the concentration of 10 μg/ml. Moreover, in this method and in the corresponding result (3.3. Inhibition of GB Cell Adhesion and Migration, lines 328-329), the Authors describe the addition of antibodies (20 μg/mL) “…to cause the collagen gel with spheroids to
float” in the methods, and “….to cover the gel”, in the Results, however, the reason is unclear. They should specify, please.

Reply: We have included reference, number 33 and 34. We have clarified in the Materials & Methods and Result sections that cell spheroids were pre-treated with integrin α10 and control antibodies and that medium containing 10 μg/ml (20 μg/ml was typing mistake and we have corrected the concentration in the main text and Materials and Methods) of the different antibodies was added to the polymerized gel (containing the spheres) to make the gel float in the antibody solution to optimize the effect of the antibodies on the cell migration.

Paragraph 2.6. BrdU Cell Cycle – The method reported is not for cell cycle analysis but for proliferation and is relative to the proliferation’ sphere and not adherent cell proliferation; the Authors should provide explanations.

Reply: Figure 5A. C2C12α10 were culture as monolayer and proliferation was performed on monolayer cells. To clarify this, we have included “Monolayer cell proliferation” as a title for this Figure and Materials and Methods.
In figures 5B and C, GB cells were cultured as spheres. To clarify, we have added the title “Sphere cell proliferation” in the Figures and Materials and Methods.

Section Results:

In the figure 2C, the representative immunofluorescent staining of integrin α10 in a GB tumor derived from U3054MG cells and grown in vivo, should be moved in the description of the in vivo results, for greater uniformity.

Reply: We prefer to show the immunofluorescent staining of a GB tumor as Figure 2C since it represents a pre-study aimed to demonstrate that integrin α10β1 is highly expressed on the U3054MG cells when grown as a tumor in vivo.

The Authors should clarify and justify the choice of some specific cell types for some experiments and the other types for other analyses. For example, why have the Authors evaluated the proliferation in the U3046MG and GBM-37 spheres and not for the U3054MG sphere? Why have authors not analyzed the
proliferation in the adherent cells?

Reply: Cells isolated from glioblastoma tissues represent a very good tool to investigate the role of integrin α10β1 in the growth of a glioblastoma tumor and in the mechanism of actions coupled to this integrin. Thus, our focus in the study is more on the integrin α10β1 as a therapeutic target and the effect of antibodies targeting this integrin than the specific cell line used. Since U3054 is isolated from a glioblastoma that has recured in the patient and likely has been extra resistant to available treatments (surgery, radiation and chemotherapy) we have found this cell line of great interest to study in vivo. However, mechanism of action studies in vitro using this cell line has been difficult for technical reasons since expression of integrin α10β1 on U3054 is low in monolayer cultures and that it proliferates and migrates very fast in 3D cultures. We have therefor in addition to U3054, used other glioblastoma cell lines as well as the integrin α10-expressing C2C12 cells to investigate the role of integrin α10 in different cell functions such as proliferation and migration. To investigate the effect of the antibodies on proliferation of adherent cells as a comparison to sphere cultures we used the C2C12α10 cells (Figure 5A).

The title of the 3.4 paragraph: Inhibition of GB Cell Proliferation and Sphere Formation, indicates the evaluation of proliferation of GB cell and not of sphere, but the data report the proliferation of spheres. Moreover two different treatment time are reported, 8 and 10 days. The Authors should clarify these
observations.

Reply: We have investigated proliferation by measuring BrdU incorporation in monolayer and in spheres as demonstrated in Figures 5A-C. As a complement we have analysed the ability of the GB cells to form spheres by analysing the number of spheres with a size over 100 μm (Figure 5D). Cell proliferation was studied after 10 days and sphere formation after 8 days as described in Materials and Methods.

Section Discussion:

In the Introduction or Discussion, the authors should comment on the reason for choosing to study the isoform α10 and not α2, which is highly expressed in all cell models.

Reply: The reason to compare expression of integrin α10β1 to the other collagen binding integrins α1β1, α2β1 and α11β1 on GB cells was to better understand the specific role of integrin α10β1 and possible involvement of the other collagen binding integrins in the adhesion and migration assays.
We have now described this in the results sections. We also wanted to investigate if 3D cultures could affect expression levels of these integrins since α1β1 and α11β1 show low expression in monolayer. No major effect was found. The integrin α2 subunit, on the other hand, was highly expressed both in monolayer and 3D cultures of the GB cells. However, this integrin is not the scope of our study. In the Discussion section we discuss the findings and refer to work from others showing that antibodies to integrin α2β1 block migration but not proliferation of glioblastoma cells.

In the Discussion, the Authors state: “However, we have shown that expression of integrin α10 largely increases on U3054MG cells grown in spheres compared to monolayer cultures [22]. Here, we confirm these results and additionally show that integrin α10 is highly expressed on the U3054MG cells grown as a
tumor in mice. The U3054MG cell line was established from a patient GB tumor that recurred after the patient had undergone surgery and radiotherapy (50.4 Gy in total), and is classified as a mesenchymal GB subtype [32] with high invasion capacity [36]. We therefore decided to use this aggressive cell line in the in
vivo study.”

In my opinion these reasons are not completely accurate. In fact, the Authors didn’t use spheres to induce glioblastoma in vivo, we do not know if using the U3046, the expression of α10 in vivo would have been as high as with the U3054. Moreover, the Authors should specify the molecular subtype of U3046.

Reply: We used U3054MG in the in vivo study since it come from a recurrent tumor and is therefore extra aggressive and difficult to treat. We assume a similar effect on U3046MG since it also expresses high integrin α10β1 on cells in spheres. We have shown that monolayer-expanded U3054MG cells form a tumor with high expression of integrin α10β1 when transplanted subcutaneously in mice (Figure 2C). Another reason to use U3054MG is that the same cell line was used in our previous study when we investigated the effect of an ADC in an orthotopic animal model. The cell line U3046MG as well as U3078MG are also classified as a Mesenchymal subtype (Xie Yet al. 2015). We have included this information in Materials and Methods Paragraph 2.1. Cell Lines and Reagent.

Finally, the authors should comment on whether the evaluation of α10 expression in the patients' tumor could guide the choice of therapy. The choice of the U3054 line would make this opportunity fall.

Reply: We have previously reported that integrin α10β1 is highly expressed in patient glioblastoma tissues and that the expression is gradually increased with increasing glioma grade both on the protein level and on gene expression level (Thorén et al. 2019). Also, increased integrin α10 gene expression correlates with lower overall survival of glioma patients (Thorén et al. 2019). Thus, high expression of integrin α10β1 in glioma tumors, could suggest a treatment strategy targeting integrin α10β1.

We have shown that integrin α10β1 on U3054 increases in 3D sphere cultures and also on monolayer expanded U3054 that were grown as a tumor in vivo (Figure 2C). Thus, the low expression of integrin α10β1 on U3054 in monolayer cultures does not represent the expression level when the cells are grown as a tumor or the expression of integrin α10β1 in patients’ GB tumors.

Minor revisions

Do the terms α10 and α10β1 indicate the same integrin? The authors should clarify this doubt, and if these terms express the same molecule, the authors should ever use the single way.

Reply: The integrin subunit α10 exist on the cell surface together with the integrin subunit β1. When we talk about the receptor, we use the full name, integrin α10β1. We have corrected in the text to be more consequent. The antibodies we use in this study have been raised against the integrin α10
subunit and we thus define them as integrin α10 antibodies although their effect is on the entire receptor, integrin α10β1.

Round 2

Reviewer 1 Report

In the present study, we have demonstrated for the first time that function blocking integrin α10- antibodies inhibit migration and proliferation of glioblastoma (GB) cells and reduce GB tumor growth in an animal subcutaneous model.………………

I would suggest inserting answer above (or something similar) you provided to my general comment, in the Discussion

However, we have shown specific immunoprecipitation of the α10 and β1 chains using C2C12α10 and C2C12α11 cells and the same results are found with a polyclonal antibody directed to the cytoplasmic domain of the human integrin α10. This polyclonal antibody detects integrin α10 in Western blot.

Where did you show specific immunoprecipitation and western? I cannot find it in the paper

Author Response

Response to Reviewer 1 Comments
Comments and Suggestions for Authors
In the present study, we have demonstrated for the first time that function blocking integrin α10- antibodies inhibit migration and proliferation of glioblastoma (GB) cells and reduce GB tumor growth in an animal subcutaneous model.………………
I would suggest inserting answer above (or something similar) you provided to my general comment, in the Discussion

We have added to the discussion that the function blocking antibodies used in this study represent a different antibody modality and have different mechanism of action compared to the ADC used in the previous study (last paragraph in discussion).

In the conclusion we write that: We have demonstrated for the first time that function blocking integrin α10-antibodies inhibit GB tumour growth as well as the migration of GB cells.

However, we have shown specific immunoprecipitation of the α10 and β1 chains using C2C12α10 and C2C12α11 cells and the same results are found with a polyclonal antibody directed to the cytoplasmic domain of the human integrin α10. This polyclonal antibody detects integrin α10 in Western blot.

Where did you show specific immunoprecipitation and western? I cannot find it in the paper

We have included a figure in the manuscript (Figure 1B) showing the binding of the integrin 10 antibodies Th101 and mAb10 to C2C1210 and C2C1211 cells and how the gating is made.
In addition, to provide the reviewer with further information about the specificity of the two monoclonal integrin α10 antibodies used in this study we below show relevant figures and also information on where results have been published.
We have over the years developed and worked with different antibodies to the integrin 10 subunit. A rabbit polyclonal antibody, raised and affinity purified against the cytoplasmic domain of human integrin 10, has been used in different assays including Immunoprecipitation (Camper et al 1998), Western Blot (Bengtsson et al 2001) and Immunohistochemistry (IHC) (Camper et al 2001). The specificity is further confirmed in the immunostaining figure below (from Bengtsson et al 2005) showing antibody staining of chondrocytes in tissue sections from a wildtype mouse and lack of staining of chondrocytes in a 10 knockout mouse. The human antibody Th101, used in the present study, show the same IHC results, binding to integrin α10 on wild type chondrocytes in mouse cartilage but not to chondrocytes from integrin 10 KO mice (figure below, unpublished data).
Further, below we also show immunoprecipitation of integrin α10β1 from Biotin-labelled C2C12α10, comparing the polyclonal antibody and the monoclonal antibody mAbα10 used in the present study. The figure shows the same results, specific immunoprecipitation of the α10 and the β1 integrin subunits from the C2C12α10 cells but not from the C2C12α11 cells. As a control, a polyclonal antibody raised against the cytoplasmic domain of α11 show immunoprecipitation of α11 and β1 from C2C12α11 cells but not from the C2C12α10. These results were part of a thesis published 2005 at Lund University (see below).

New figure 1B added to the manuscript.

(please see in attached file)

Figure legend. Specific binding of Th101 and mAba10 to C2C12 cells transfected with the human integrin subunit α10. C2C12α10 and C2C12α11 cells were stained with primary antibodies for 30 minutes in the dark at 4°C, washed twice with FACS buffer (DPBS containing 1% FBS and 0.1% sodium azide), incubated with secondary antibody for 30 minutes at 4°C and then washed twice with FACS buffer and analyzed using a BD Accuri C6 flow cytometer. The gates were set through unstained cell samples. The figure shows clear cell plot shift when the
C2C12α10 cells were stained with the integrin α10 antibodies human Th101 and mouse mAbα10 antibodies, but not with the human isotype control antibody Th301, the mouse isotype control antibody IgG2a or integrin α11 mouse monoclonal antibody (mAbα11). In C2C12α11 cells, there was cell plot shift when the cells were stained with mAbα11, but not with Th101 and mAbα10.

These results demonstrate that the integrin α10 antibodies Th101 and mAbα10 specifically bind to integrin α10 on C2C12 cells and do not cross react with C2C12 cells expressing integrin α11.

Results from previous work:
Immunostaining with polyclonal α10 antibody (from Bengtsson et al. 2005). Please, see the reference section below.

(see figure in attached file)

Figure legend. Immunostaining of chondrocytes in cartilage from newborn wild-type and mutant mice using the polyclonal α10 integrin. The results show binding to the wild type chondrocytes expressing integrin the α10 subunit
but not to mutant chondrocytes lacking α10 integrin.

Immunostaining with the human monoclonal Th101 (Unpublished results, from an ongoing study).

(see figure in attached file)

Figure legend. Immunohistochemical staining of chondrocytes in cartilage from 14 days old wild-type and knockout (mutant) mice using the α10 integrin monoclonal antibody Th101 and the control antibody Th301, both used in
the present study. The results show specific binding of Th101 to the wild type chondrocytes expressing integrin the 10 subunit similar to the polyclonal antibody above.

Immunoprecipitation experiments comparing the monoclonal mAbα10 with the polyclonal integrin α10 antibody (From PhD thesis by Lars Bryngelsson Ohlsson, Lund University 2005).

(see figure in attached file)

Figure legend. C2C12α10 and C2C12α11 cells were biotinylated to label cell surface proteins and cell lysates were then immunoprecipitated with either the polyclonal integrin α10 antibody, a polyclonal integrin α11 antibody
or the monoclonal integrin α10 antibody mAbα10. Lanes: 1-3 show IP of C2C12α10 cells; lanes 4-6 show IP of C2C12 α11 cells. Lanes 1 and 4 = Polyclonal α10 antibody; lanes 2 and 5: Polyclonal α11 antibody; lanes 3 and 6: mAb α10

References:
Camper L, Hellman U, Lundgren-Akerlund E. Isolation, cloning, and sequence analysis of the integrin subunit alpha10, a beta1-associated collagen binding integrin expressed on chondrocytes. J Biol Chem. 1998; 273(32):20383-9.
Camper L., Holmvall K. Wängnerud C., Aszódi A. Lundgren-Åkerlund E. Distribution of the collagen-binding integrin α10β1 during mouse development. Cell Tissue Res 2001; 306:107–116.
Bengtsson T, Camper L, Schneller M, Lundgren-Akerlund E. Characterization of the mouse integrin subunit alpha10 gene and comparison with its human homologue. Genomic structure, chromosomal localization and identification of splice variants. Matrix Biol. 2001; 20(8):565-76.
Bengtsson T, Aszodi A, Nicolae C, Hunziker BE., Lundgren-Akerlund E, Fässler R. Loss of α10β1 integrin expression leads to moderate dysfunction of growth plate chondrocytes. Journal of Cell Science 2005; 118, 929-936.
Lars Bryngelsson Ohlsson Studies of Mesenchymal Progenitor Cells and Tumour Growth, Integrins and Matrix Metalloproteinases. 2005 PhD thesis; Publication in Lund University research portal; Department of Experimental Medical Science, Lund University.

Reviewer 2 Report

The manuscript was well revised in accordance with reviewer’s comments.

Author Response

"The manuscript was well revised in accordance with reviewer’s comments". No attachment required. 

We want to thank the reviewer for the revision of our manuscript.

Round 3

Reviewer 1 Report

accept in the revised form